# Pharmacology of *Veratrum californicum* Alkaloids as Hedgehog Pathway Antagonists

**DOI:** 10.3390/ph17010123

**Published:** 2024-01-17

**Authors:** Madison L. Dirks, Owen M. McDougal

**Affiliations:** 1Biomolecular Sciences Graduate Programs, Boise State University, Boise, ID 83725, USA; madisondirks@u.boisestate.edu; 2Department of Chemistry and Biochemistry, Boise State University, Boise, ID 83725, USA

**Keywords:** *Veratrum californicum*, alkaloid, hedgehog signaling pathway, bioactivity, cyclopamine, cancer, basal cell carcinoma

## Abstract

*Veratrum californicum* contains steroidal alkaloids that function as inhibitors of hedgehog (Hh) signaling, a pathway involved in the growth and differentiation of cells and normal tissue development. This same Hh pathway is abnormally active for cell proliferation in more than 20 types of cancer. In this current study, alkaloids have been extracted from the root and rhizome of *V. californicum*, followed by their separation into five fractions using high performance liquid chromatography. Mass spectrometry was used to identify the presence of twenty-five alkaloids, nine more than are commonly cited in literature reports, and the Bruker Compass Data Analysis software was used to predict the molecular formula for every detected alkaloid. The Gli activity of the raw extract and each fraction were compared to 0.1 µM cyclopamine, and fractions 1, 2, and 4 showed increased bioactivity through suppression of the Hh signaling pathway. Fractions 2 and 4 had enhanced bioactivity, but fraction 1 was most effective in inhibiting Hh signaling. The composition of fraction 1 consisted of veratrosine, cycloposine, and potential isomers of each.

## 1. Introduction

Few treatments exist outside of surgery for basal cell carcinomas (BCCs), and the drug therapies which are available are considered a last resort [1,2]. Patients often discontinue drug therapy prematurely due to extreme side effects including alopecia, fatigue, muscle spasms, and others [1,2]. An under-explored source of potential treatment exists in the high mountain meadows of Idaho. *Veratrum californicum* contains steroidal alkaloids that are potent hedgehog (Hh) signaling inhibitors [3]. The Hh signaling pathway is implicated in the growth of at least 20 types of cancer, and drug treatment has recently focused on targeting this pathway [4]. Cyclopamine, the most well-studied alkaloid extracted from *V. californicum*, has been used to understand the mechanism of the Hh signaling pathway in cancer progression and has served as a molecular scaffold for modern chemotherapeutics [2,5,6,7]. Early studies of *V. californicum* only examined major components of the root and rhizome raw extract [8,9,10,11,12]. In this current study, we have turned our attention to less abundant alkaloids in the *V. californicum* root and rhizome in order to uncover molecules that suppress Hh signaling. Herein, we present two previously unidentified alkaloids that have been detected and demonstrate potent Hh signaling suppression. The potential isomers of veratrosine and cycloposine warrant further examination as small molecule antagonists of Hh signaling that may be pursued for drug discovery research.

### 1.1. Veratrum Californicum Alkaloids

During the 1950s, in the mountain meadows of Idaho, sheepherders reported that up to a quarter of their newborn sheep had a craniofacial malformation [5,13]. Deformities affecting their skulls, jaws, sometimes brains, and eyes led to the term “monkey-faced” lambs [13]. The most characteristic feature was a singular, enlarged, cyclopean eye in the middle of the face [13]. In 1954 the Poisonous Plant Research Laboratory (PPRL) in Logan, Utah, was given the task of discovering the origin of these mutations [5]. At the time the cause of the abnormalities was proposed to be a recessive genetic trait, but a breeding study in 1957 eliminated this possibility [13]. The next step in determining the source was performing field and feeding studies [5].

*V. californicum* was explored as a potential cause of the lambs’ alterations in 1958 after a sheepherder observed it causing the sheep to become sick following consumption. Feeding trials began soon after, and the PPRL reported the sheep experiencing a variety of infirmities, including excessive salivation and frothing at the mouth, vomiting, abnormal gate, irregular heartbeat, dyspnea, convulsions, coma, death, and in 1959 a cyclopean eye was produced. After several more years of trials, a definitive correlation was made between the ewes’ consumption of *V. californicum* on day 14 of gestation to lambs born with cyclopean malformations underneath a proboscis-like nose [3,5].

In the search for the molecular explanation of these symptoms, the alkaloid cyclopamine was separated from *V. californicum* raw extract in 1968, and in 1969 its steroidal structure was published [10,11]. Cyclopamine had been previously identified and called 11-deoxojervine in Japan in 1965 after its initial discovery in *V. californicum*’s sister plant *Veratrum grandiflorum* [5]. It was concluded that the three alkaloids cyclopamine, cycloposine, and jervine were responsible for the lambs’ deformities. Previously, other *Veratrum* alkaloids (cevanine-type) had been studied for their hypotensive properties, but it became clear by 1970 that some of these newer discoveries were teratogenic [3]. *V. californicum* alkaloid discovery continued for a few more years, and then remained relatively untouched until the early 2000s.

Eventually more precise instrumentation and techniques became available for solid phase extraction, which greatly improved the extraction, separation, isolation, characterization, and bioactivity assessment of *V. californicum* alkaloids [14]. Over the past decade, our lab has explored less abundant alkaloids from *V. californicum* that were under-studied [15]. This work began with a comparison of eight methods for extracting cyclopamine from *V. californicum*, which led to the conclusion that ethanol extraction produced the highest number of alkaloids, with the greatest retained bioactivity [16]. Next, cyclopamine, veratramine, muldamine, and isorubijervine were quantified in the different plant parts: roots/rhizomes, stems, and leaves [17]. Additionally, the variation in alkaloid content was explored for the various plant parts, stages of growth, and harvest locations. This led to the detection of six new uncharacterized alkaloids, all of which appeared to exhibit Hh pathway suppression activity, thus inspiring this current work [4].

### 1.2. Hedgehog Signaling Pathway

*V. californicum* alkaloids are of interest due to their inhibition of the Hh signaling pathway, which is critical for intercellular communication during fetal development [18,19,20]. The Hh pathway involves proper cell polarization, most epithelial tissue differentiation leading to bilateral symmetry, and the formation of the central nervous system, skeleton, limbs, teeth, eyes, and several vital organs within vertebrate embryos [4,5,19,21,22,23,24]. Loss of the pathway’s proper function can result in a range of complications from under-developed facial features to cyclopia to nervous system disorders [5]. In addition, propagation of over 20 cancers has been correlated to aberrant Hh signaling. Blocking the Hh signaling pathway during development can be detrimental, but inhibition in cancer is a treatment option [3,4].

The Hh gene was first discovered in 1978 through *Drosophila* gene knockout trials conducted by Christiane Nusslein-Volhard and Eric Wieschaus, who later earned a Nobel Prize for their work. When the Hh gene was silenced, the fruit fly embryo was short with spine-like projections reminiscent of a hedgehog [5,25]. The Hh pathway in vertebrates utilizes three signaling molecules, which were named after hedgehog breeds and a fictional character: Desert hedgehog (Dhh), Indian hedgehog (Ihh), and Sonic hedgehog (Shh) [5]. These molecules must be secreted with the help of the membrane protein Dispatched (DISP1), cholesterol, and secreted protein SCUBE2 (Figure 1, 1.) [26,27]. When Hh protein binds Patched 1 (PTCH)—a 12-span transmembrane protein—inhibition of Smoothened (SMO)—a seven-span transmembrane protein—is interrupted, and a signaling cascade is initiated (Figure 1, 2.) [5,18,26,27,28,29,30,31,32]. SMO is phosphorylated using GRK2 and casein kinase 1 (CK1) and begins to collect in the primary cilium (Figure 1, 3.) [26,28]. This causes the inhibition of Suppressor of Fused (SuFu) and Kif7, which then release a glioma associated oncogene homolog (Gli) (Figure 1, 4.) [26,28,29,30]. The levels of zinc-finger transcription factors from the cubitus interruptus (Ci)/Gli family determine if the Hh signal is transmitted [5,18]. Gli 1 and Gli 2 function as transcriptional activators (Figure 1, 5.), while Gli 3 is a repressor [5,18,26,28,29,30,31]. When the pathway is off, PTCH inhibits SMO, allowing SuFu and Kif7 to sequester Gli (Figure 1, 6. and 7.) [26,28,29,30,31]. SMO is marked for degradation, and SuFu and Gli are phosphorylated using protein kinase A (PKA), glycogen synthase kinase 3 (GSK3), or CK1 (Figure 1, 8. and 9.) [26,29,30,32]. Gli can then take one of two paths, which are mediated using β-TRCP, an E3 ubiquitin ligase, and its adaptor protein, SPOP [28]. First, it can be marked for degradation and passed onto the proteosome, and second, it can be proteolytically cleaved and enter the nucleus as Gli3, inhibiting Hh gene transcription (Figure 1, 10. and 11.) [30,31]. Another method of inhibiting transcription is by using the teratogenic alkaloid cyclopamine, the first discovered Hh pathway suppressor, which replaces PTCH, stopping signal transduction [5,18].

Cyclopamine has been used to inhibit the Hh signaling pathway when it is irregularly activated by cancer [3,5,33]. In mice, cyclopamine has suppressed cancerous tumor growth in several models, including xenografts models of human colon cancer [34], glioma [35], melanoma [36], pancreatic cancer [37,38,39], prostate cancer [40], and small cell lung cancer [41], in addition to a medulloblastoma allograft [42], and a genetic medulloblastoma model [5,43]. In humans, cyclopamine has been infused into an oil-based ointment for topical application directly to tumors, which has led to tumor size reduction without detrimental side effects [5].

As a drug therapy, cyclopamine does have some weaknesses: it is not well solubilized in water or physiological environments, it degrades in acidic environments (e.g., stomach acid), and it can inhibit cellular neurogenesis and proliferation as adult neural stem cells still utilize the Hh pathway into maturity [4,5,33,44,45]. Exploration for new Hh inhibitor therapies began with the structure of cyclopamine as a prototype. This work started at Johns Hopkins University, where a 3-keto *N*-(aminoethyl-aminocaproyl-dihydrocinnamoyl) moiety was combined with cyclopamine (KAAD-cyclopamine) to increase its solubility, and in mouse models the new molecule was 10–20 times more potent than cyclopamine alone [5]. Further studies based on cyclopamine’s molecular scaffold were conducted to see if the side effects could be reduced, which resulted in IPI-926 (a.k.a. patidegib), a semi-synthetic variant with increased stability and potency [7]. Phase 3 clinical trials have been completed for IPI-926 in a 2% topical gel for the treatment of basal cell carcinomas for patients with basal cell nevus syndrome (Gorlin syndrome), but the results were not reported [46]. Computational research modeled off cyclopamine was performed to identify additional molecules of interest. In one instance, a multitude of small molecules were screened for their ability to bind SMO [6]. In 2012, FDA approval was given for Vismodegib (GDC-0449 or Erivedge) to treat metastatic basal cell carcinoma (mBCC) or locally advanced basal cell carcinoma (laBCC) in adults unable to undergo surgery or radiation therapy [6,47]. Developed by Genentech, Inc., Vismodegib had an overall response rate (ORR) of 60% in patients with laBCC and 46% in patients with mBCC in Phase II clinical trials. In this study, 30% of mBCC patients demonstrated a decrease in tumor size, and 43% of laBCC patients displayed lesions healing or a significant decrease in tumor size. Common side effects included decreased appetite, diarrhea, dysgeusia, fatigue, hair loss, muscle spasms, and nausea [6]. A small-molecule in vitro screening performed by Sun Pharma Global resulted in Sonidegib (LDE225, erismodegib, or ODOMZO) receiving FDA approval in 2015 to treat recurrent advanced basal cell carcinoma (aBCC) in patients unable to undergo surgery or radiation [2,33,48]. In Phase II clinical trials, after 12 months of treatment, patients with laBCC (18 of 94) either died or had their condition progress, but the ORR was 57.6% with 200 mg and 43.8% with 800 mg. For patients with mBCC the ORR was 7.7% for 200 mg and 17.4% for 800 mg [49]. From most to least common, the potential side effects of taking Sonidegib were muscle spasms, alopecia, dysgeusia, nausea, increased creatinine kinase, fatigue, weight loss, diarrhea, decreased appetite, myalgia, and vomiting [2]. Eventually, patients experienced resistance to both Vismodegib and Sonidegib as a result of a SMO mutation or the initiation of an alternative Hh pathway [50]. In 2018, Pfizer’s Glasdegib (PF-0449913 or DAURISMO^™^) in combination with low-dose cytarabine (LDAC) was approved by the FDA as an oral treatment for acute myeloid leukemia (AML) patients of at least 75 years of age or patients with co-morbidities who cannot undergo intensive induction chemotherapy [33,51]. In the BRIGHT AML 1003 study, when contrasted with only LDAC, Glasdegib paired with LDAC demonstrated a decrease in mortality by 54%, but patients experienced pneumonia, fatigue, dyspnea, hyponatremia, sepsis, and syncope [51]. The molecular structures for each of the alkaloids (cyclopamine, KAAD-cyclopamine, and IPI-926) and chemotherapeutics (Vismodegib, Sonidegib, and Glasdegib) can be seen in Figure 2.

With cyclopamine being the first identified molecule with the ability to block Hh signaling and cyclopamine variants exhibiting increased Hh signaling suppression, it is reasonable to revisit *V. californicum* in the quest for more inhibitors that may have been overlooked due to their low natural abundance at the time of the original work in the 1950s. Preliminary work in our lab resulted in the detection of sixteen alkaloids, where six were identified using commercially available standards, and the identities of five others were speculated based on the predicted molecular formula obtained using mass spectrometry (MS) from the mass to charge (*m*/*z*) ratio [4,17]. The bioactivity work demonstrated that the raw root/rhizome extract was more effective at Hh signaling inhibition than a proportionate amount of cyclopamine [4,52].

The purpose of this current work was to detect and evaluate the minor constituents present in the raw extract that inhibit Hh signaling. The experimental approach was to collect different fractions of the raw root/rhizome ethanolic extract to focus on alkaloid abundant fractions with the highest level of bioactivity as measured using Shh-Light II cell assay signal suppression [53]. Five fractions of alkaloids were collected at the retention times of 10.75–13.25, 13.25–15.75, 15.75–18.25, 18.25–20.75, and 20.75–23.25 min from the raw extract chromatogram using high performance liquid chromatography (HPLC) with a diode array detector (DAD) and an automated fraction collector (see Section 3). Then, a cyclopamine standard, the raw extract, and the five fractions were tested via measured Gli activity on a Johns Hopkins University Shh-Light II cell line (JHU-068). Three fractions were statistically significantly superior in Hh signaling suppression when contrasted with the cyclopamine standard. Fractions 1, 2, and 4 should be pursued further to isolate and examine additional *V. californicum* alkaloids with the potential to be used as cancer therapies. Fraction 1 is of particular interest, as it had by far the greatest Hh inhibition with only four alkaloids present, two of which have been studied previously and have been reported to not suppress Hh pathway signaling in this model system, implying the remaining two alkaloids are responsible for the observed bioactivity. Nuclear magnetic resonance and high-resolution mass spectrometry (HRMS) experiments were conducted for these two alkaloids, resulting in their potential identification as isomers of cyclopamine and veratramine [4,52].

## 2. Results and Discussion

### 2.1. Extraction and Fraction Collection

Previous work in our lab was performed using similar methods of ethanol extraction and HPLC-MS characterization, where peaks of interest were obtained on an HPLC chromatogram between 12.8 and 24.5 min [16,17]. With an adjustment to the concentration gradient for this work, the alkaloids were observed to elute earlier (10.75–23.25 min). When the raw extract was examined using the charged aerosol detector (CAD), alkaloid peaks were resolved between 10 and 20 min (Appendix A). Five fractions were collected in 150 s intervals beginning at 10.75 min and ending at 23.25 min. These fractions were collected from several runs, pooled, dried, and reconstituted in order to concentrate the alkaloids for enhanced detection. Each fraction contained between three and ten peaks of interest (Figure 3). Chromatographic analysis and MS were used to identify and further characterize the alkaloids present.

Each fraction was analyzed using high resolution CAD (Appendix A) and MS to determine the molecular weight and predicted molecular formula based on the *m*/*z* ratio. The mass spectra and extracted ion chromatograms of abundant alkaloids from the five fractions are shown in Appendix A.

Within the five fractions there were twenty-five unique molecules detected, sixteen of which were found in only one fraction and nine of which were found in at least two fractions. Identification of cyclopamine and veratramine was established by co-elution with commercially available standards (Appendix A), the identities of sixteen alkaloids were suspected based on a literature precedent, and seven alkaloids did not correlate with the literature. Table 1 shows the alkaloids that were identified, the m/z ratio, and the predicted molecular formula.

Examination of the chromatograms corresponding to fractions 1–5 (Appendix A) reveals four alkaloids in fraction 1, thirteen (including those from Appendix A) in fraction 2, three in fraction 3, seven (including one from Appendix A) in fraction 4, and eight in fraction 5. MS was used to identify veratrosine, cycloposine, and a potential isomer of each in fraction 1 (Appendix A). Cycloposine and a potential isomer, a potential isomer of veratrosine, tetrahydrojervine, dihydrojervine and a potential isomer, etioline, veratramine, and five unknown alkaloids were found in Fraction 2 (Appendix A). Fraction 3 contained veratramine, cyclopamine, and a potential isomer of cyclopamine, which were all observed in other fractions (Appendix A). Fraction 4 contained isorubijervine, a potential isomer of veratramine, cyclopamine and a potential isomer, muldamine, and two unknown compounds (Appendix A). Finally, cyclopamine, muldamine and a potential isomer, 22-keto-26-aminocholesterol and a potential isomer, verazine, and two unknown compounds were present in fraction 5 (Appendix A) [4].

Etioline, dihydrojervine, and isorubijervine had been detected in *V. californicum* in the past, but in this work, they were not visible in the HPLC chromatograms. MS was used to identify these compounds using retention times, *m*/*z* ratios, and predicted molecular formulas corresponding to data from prior reports (see Appendix A) [4]. The validation of veratramine and cyclopamine by co-elution with standards is shown in Appendix A.

### 2.2. Bioactivity

The Dual-Luciferase^®^ Reporter Assay System was used to report the Shh-Light II cells’ bioactivity [53]. Hh signaling was initiated by the addition of Shh protein, which caused an influx of Gli transcription factors. A plate reader recorded the luminescence, which functioned as a measure of Hh pathway activation [5,18]. The 96-well plates had three wells for treatment with negative control (media only), positive control (media and Shh protein), high concentration (0.1 µM) cyclopamine or low concentration (0.01 or 0.05 µM) cyclopamine, 1:200 dilution of raw alkaloid extract, 1:1000 dilution of raw alkaloid extract, 1:20 dilution of each fraction, and 1:100 dilution of each fraction in media with added Shh protein. Figure 4a,b displays the averaged findings from the four plates.

In Figure 4a, the relative Gli-reporter activity for cyclopamine (final concentration of 0.1 µM) was 63.56 ± 18.40, while fraction 1 was 1.38 ± 8.23, fraction 2 was 12.04 ± 11.66, and fraction 4 was 17.73 ± 9.10. Fractions 2 and 4 had significant (*p* < 0.05) Hh suppression, but fraction 1 was even more potent with *p* < 0.01 significance. Figure 4b did not have any significant Hh pathway inhibition. Most of the samples had a large margin of error, including the controls, where the negative control had a relative Gli-reporter activity of 33.97 ± 34.32, and the positive control had 155.19 ± 54.25. The bioactivity results are based on the relative comparison of qualitative experiments, which we believe attributed to the relatively high variation observed.

Alkaloids present in fractions 1, 2, and 4 displayed better Gli-reporter inhibition than 0.1 µM cyclopamine. The abundance of each of the alkaloids identified in these three fractions was explored further. Figure 5 used the peak areas in the total ion chromatograms to compare the relative concentrations of alkaloids in fractions 1, 2, and 4 to 0.1 µM cyclopamine.

In Figure 5a the total relative alkaloid content can be seen to be far lower for 0.1 µM cyclopamine compared to fractions 1, 2, and 4. Even with only four alkaloids (cycloposine, veratrosine, and a potential isomer of each), fraction 1 had the lowest Gli-reporter activity, and thus best Hh inhibition. There were at least thirteen different alkaloids in fraction 2, but the combination of fraction 2 alkaloids was not nearly as potent as the combined alkaloids from fraction 1. In Figure 5b, a large amount of cyclopamine can be seen in fraction 4, but it was not nearly as efficient at suppressing Hh signaling as fraction 1. Keeler proposed the digestion of the glycosylated alkaloids produced teratogenic effects within the sheep, but cycloposine and veratrosine from fraction 1 do not exhibit Hh inhibition in this system [3,12,17]. The resulting bioactivity must originate from the potential isomers of cycloposine and veratrosine, which are yet to be characterized and have not been reported previously in the literature. The potential isomer of veratrosine had an elution time of 16.0 min, an *m*/*z* of 572.3, and a predicted molecular formula of C_33_H_49_NO_7_, and the potential isomer of cycloposine had an elution time of 16.5 min, an *m*/*z* of 574.3, and a predicted molecular formula of C_33_H_51_NO_7_. Because of the potency of fraction 1 and its low relative alkaloid concentration, these two alkaloids appear to be suitable targets for further characterization. Figure 5b shows the relative amounts of cyclopamine (0.767) compared to the potential isomer of veratrosine (0.565) and the potential isomer of cycloposine (1.31). Again, this is a relative comparison for a qualitative assessment of the alkaloid content. Future experiments should focus on quantifying the masses of each alkaloid present.

## 3. Materials and Methods

### 3.1. Plant Material

On 3 July 2014, *V. californicum* plants were gathered from beside the Shindig Trail in the Boise National Forest, Idaho (N 43 45.719″ W 116 05.327″). The above ground plant was discarded, and the roots/rhizomes were put on ice for transportation to the laboratory. A LabConco Freezone 4.5 freeze drying unit (Labconco Corporation, Kansas City, MO, USA) was used to freeze dry the plant parts for 14 h before storage in the freezer in sealed plastic bags [4].

### 3.2. Extraction

To begin the extraction process, plant parts were thawed at room temperature and then cut into small pieces (approximately 2 cm). Liquid nitrogen was used to refreeze them before they were returned to the same freeze–drying unit for another 24–48 h. The pieces were ground into a powder using a coffee grinder (Mr. Coffee, IDS77), and the powdered plant material was either frozen in a vacuum-sealed bag or used immediately for extraction. A 25 g portion of powdered plant material and 50.0 mL 95% ethanol were combined prior to 30 min of sonication. The solution was vacuum filtered (Whatman filter paper, 0.45 µm) after stirring on a stir plate overnight. The filtrate was collected for rotary evaporation at reduced pressure to remove the ethanol and dry the product. A volume of 10.0 mL 95% ethanol was used to resuspend the residual solid. This solution was heated in a warm water bath to approximately 40 °C followed by five min of sonication. The pH was raised to 10 or above by the addition of ammonium hydroxide before the solution was allowed to absorb for 10 min on a supported liquid extraction (SLE) column (Chem Elut, Agilent, Santa Clara, CA, USA or HyperSep SLE, ThermoFisher Scientific, Pittsburgh, PA, USA). A vacuum manifold was used to elute the alkaloids with chloroform (3 × 10 mL). These fractions were combined, and the chloroform was removed with rotary evaporation. The raw extract was composed of the remaining alkaloid residue suspended in 1.0 mL 100% ethanol.

### 3.3. Separation

A Dionex UltiMate^®^ 3000 HPLC system (Thermo Scientific, Waltham, MA, USA) paired with an automated fraction collector and a DAD was utilized for fraction collection and the initial examination of the raw extract. Separation was achieved with a semi-preparative Zorbax SB-C_18_ column (Agilent, Santa Clara, CA, USA) (9.4 × 250 mm, 5 µm) using eluents of 0.1% trifluoroacetic acid (TFA) in water (Buffer A) and HPLC grade acetonitrile (Buffer B). The method used a 3.0 mL/min flow rate and began with 15% Buffer B increasing to 60% Buffer B over a 25 min period followed by 15% Buffer B from 25.1 min to 30 min. From 10.75 min to 23.25 min, five fractions were obtained in 150 s intervals. Multiple runs were completed, and the fractions from the same time intervals were combined. Fractions 1–5 were dried with rotary evaporation, dissolved in 2.0 mL 100% ethanol, and frozen until HPLC-MS analysis was performed.

### 3.4. Identification

A Thermo Scientific UltiMate 3000 HPLC (Thermo Scientific, Waltham, MA, USA) coupled to a Corona Veo RS CAD was used for the analysis of the fractions and raw extract. Evaluation was performed with eluents of 0.1% TFA in water (Buffer A) and HPLC grade acetonitrile (Buffer B) on a Thermo Acclaim 120 C_18_ column (2.1 × 150 mm, 3 µm). The method began with 15% Buffer B at a flow rate of 0.3 mL/min, then increased linearly to 60% Buffer B over 25 min, and then returned to 15% Buffer B from 25.1 to 30 min. Cyclopamine and veratramine standards (both 10.1 mM) were used to confirm peak retention times.

Dr. Xinzhu Pu, Biomolecular Research Center manager, performed further characterization using HPLC-MS analysis with an ultra-high resolution Quadrupole Time-of-Flight (QTOF) MS (Bruker maXis). The electrospray ionization (ESI) source was operated under the following conditions: positive ion mode, 1.2 bar nebulizer pressure, 8 L/min flow of N_2_ drying gas heated to a temperature of 200 °C, 3000 V to −500 V voltage between the HV capillary and HV end-plate offset, and the quadrupole ion energy at 4.0 eV. The instrument was calibrated to a mass range of 80 to 800 *m*/*z* using sodium formate. Separation was performed with eluents of 5% acetonitrile and 0.1% formic acid in water (Buffer A) and acetonitrile and 0.1% formic acid (Buffer B) with an XTerra MS C_18_ column, 3.5 μm, 2.1 × 150 mm (Waters, Milford, MA, USA). A 1 μL injection volume with a flow rate of 200 μL/min was used with a method beginning at 5% Buffer B, which increased linearly to 70% Buffer B for 25 min. The mass and molecular formula for alkaloids were predicted using the Compass Data Analysis 4.0 software (Bruker Corporation, Billerica, MA, USA).

### 3.5. Chemicals and Solvents

The extraction solvents, ethanol (95%), ammonium hydroxide (25–30%), and chloroform, were purchased from Fisher Scientific (Pittsburgh, PA, USA), and ethanol (100%) was purchased from Decon Labs (King of Prussia, PA, USA). The HPLC mobile phases TFA and acetonitrile (>99% purity) were also obtained from Fisher Scientific. The cyclopamine standard (>99% purity) was purchased from Alfa Aesar (Ward Hill, MA, USA), and the veratramine standard (>98.0% purity) was purchased from Tokyo Chemical Industry (TCI) (Tokyo, Japan).

### 3.6. Cell Culture

The Shh-Light II cells were grown in an incubator, which maintained an atmosphere of 5% CO_2_, 100% relative humidity, and a temperature of 37 °C. The growth medium was composed of Dulbecco’s Modified Eagle Medium (DMEM) (Gibco, Carlsbad, CA, USA), 10% fetal bovine serum (FBS), and with added antibiotics geneticin (0.4 mg/mL), Zeocin™ (0.15 mg/mL from Invitrogen), and 1% penicillin-streptomycin. Bioactivity was measured using the Dual-Luciferase^®^ Reporter Assay System (Promega, Madison, WI, USA). Four 96-well plates were analyzed, where each was seeded with 10,000 cells per well in 100 µL of growth medium. The first plate was seeded after the cells’ fifth passage, the second after the eighth passage, the third after the eleventh, and the fourth was from a new stock after the second passage. After the cells had grown to 80% confluency, the medium was replaced with DMEM with added 0.5% FBS, 0.5% penicillin-streptomycin, and 2.5 or 5.0 µg N-terminal mouse recombinant Shh (R&D Systems, Minneapolis, MN, USA), and controls or treatments were applied to each well in triplicate resulting in 1% ethanol content. The Dual-Luciferase^®^ Reporter Assay System was used to lyse the cells and cause firefly luciferase luminescence through the addition of 100 µL Luciferase Assay Substrate in Luciferase Assay Buffer II (LAR II). All luminescence was measured using a BioTek Synergy H1m Microplate reader (BioTek, Winooski, VT, USA). Next, *Renilla* luciferase luminescence was induced using 100 µL Stop & Glo^®^ Substrate in Stop & Glo^®^ Buffer (Stop & Glo^®^ Reagent). The Dual-Luciferase^®^ Reporter Assay System manual described the calculation for the relative response ratio (RRR), which was used to express relative Gli activity.

## 4. Conclusions

Separating the raw extract into fractions with collection intervals of 150 s was intended to more easily assess *V. californicum* alkaloids with the most desirable bioactivity. The region of the chromatogram used for fraction collection (10.75–23.25 min) was selected for the presence of alkaloids with suspected Hh signal suppression as previously reported in the literature [17]. The alkaloid content in each fraction was observed using CAD and MS, and the bioactivity characteristics were pursued using an in vitro assay. Fractions 1, 2, and 4 were significantly more effective at Hh signaling inhibition than 0.1 µM cyclopamine. Fraction 1 exhibited the greatest Hh signaling inhibition, yet two of the four alkaloids present are not Hh signaling suppressors. It may be inferred that the potential isomers of cycloposine and veratrosine are the source of bioactivity among the alkaloids present in fraction 1. Further work is required to characterize the alkaloids with notable bioactivity. Inhibition of the Hh signaling pathway can lead to the suppression of cancer proliferation. This work provides a path toward identifying new treatments for patients suffering from BCCs and other cancers that act through aberrant Hh pathway signaling.

## Figures and Tables

**Figure 1 pharmaceuticals-17-00123-f001:**
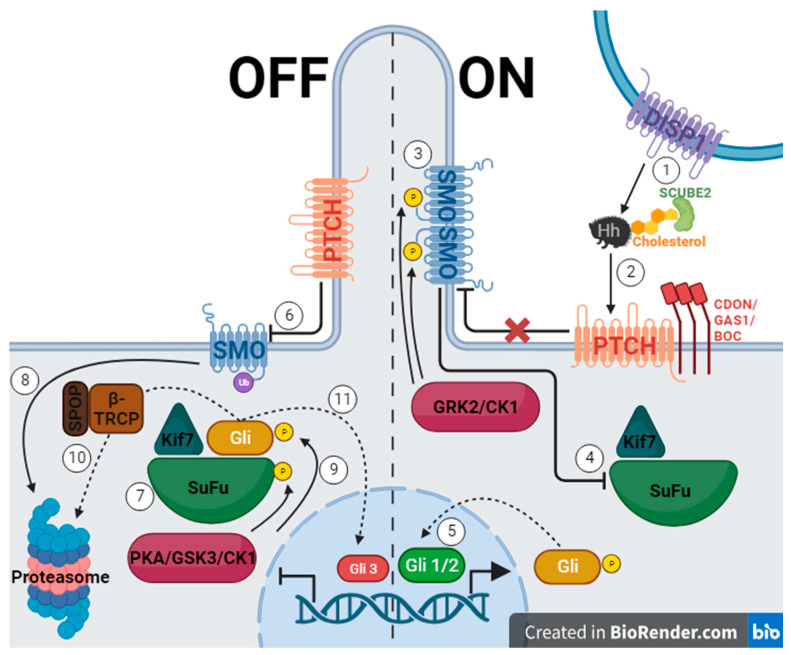
The hedgehog signaling pathway, where the left side of the figure represents the off state of the pathway due to PTCH inhibiting SMO, and the right side represents the on state when the pathway is activated by Hh protein binding to PTCH allowing SMO to function. (Created with BioRender.com).

**Figure 2 pharmaceuticals-17-00123-f002:**
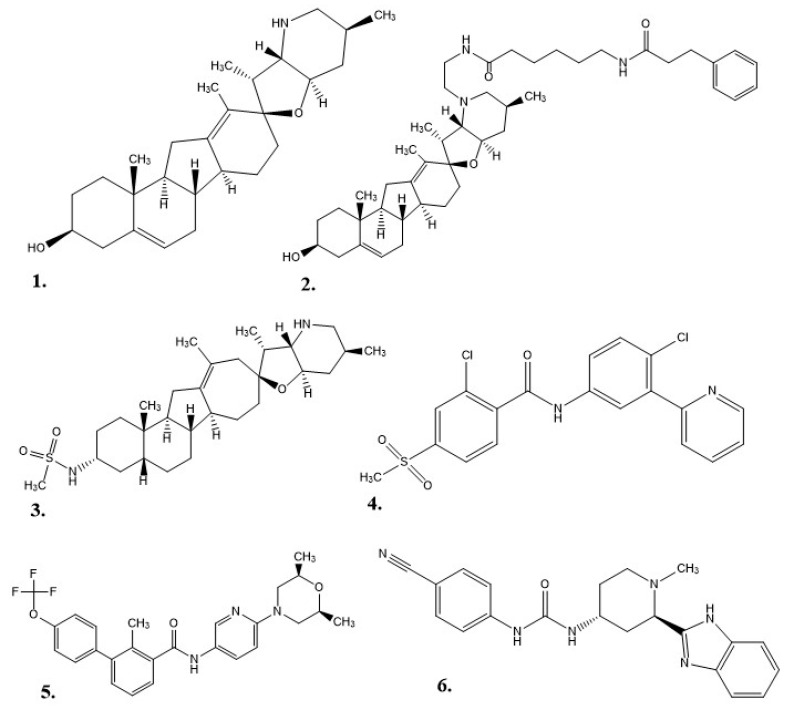
Structures of hedgehog signaling pathway inhibitors. (**1**) Cyclopamine, (**2**) KAAD-cyclopamine, (**3**) IPI-926, (**4**) Vismodegib, (**5**) Sonidegib, (**6**) Glasdegib.

**Figure 3 pharmaceuticals-17-00123-f003:**
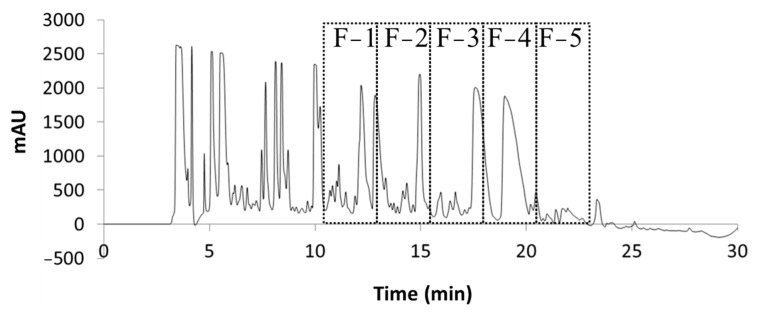
HPLC-DAD chromatogram of *V. californicum* raw extract with fractions 1–5 boxed.

**Figure 4 pharmaceuticals-17-00123-f004:**
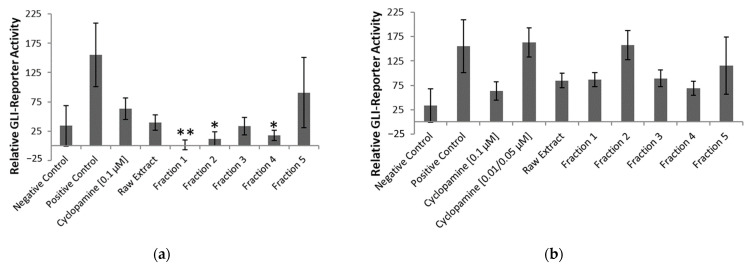
Results from the bioactivity assessment of cyclopamine, raw extract, and five fractions: (**a**) high (1:200 dilution for raw extract and 1:20 dilution for fractions) concentrations of each treatment, where * signifies *p* < 0.05 and ** signifies *p* < 0.01 as compared to 0.1 µM cyclopamine; (**b**) low (1:1000 dilution for raw extract and 1:100 dilution for fractions) concentrations of each treatment as compared to 0.1 µM cyclopamine.

**Figure 5 pharmaceuticals-17-00123-f005:**
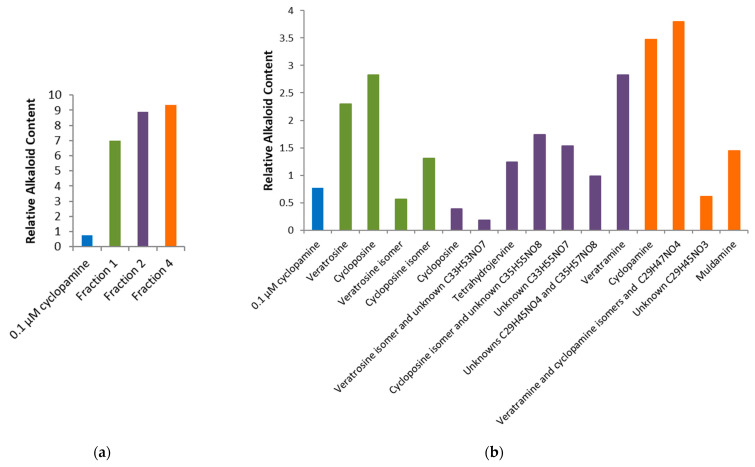
Relative concentrations of (**a**) total alkaloid content and (**b**) individual alkaloids from fractions 1, 2, and 4 as compared to 0.1 µM cyclopamine.

**Table 1 pharmaceuticals-17-00123-t001:** Summary of findings from the five fractions of *V. californicum* raw extract including retention time (R_t_), *m*/*z*, predicted molecular formula, identity, and fraction of origin.

R_t_	*m*/*z*	Molecular Formula	Identity	Fraction
15.3	572.3	C_33_H_49_NO_7_	Veratrosine	1
15.7	574.3	C_33_H_51_NO_7_	Cycloposine	1, 2
16	572.3	C_33_H_49_NO_7_	Isomer of Veratrosine *	1, 2
16.1	576.4	C_33_H_53_NO_7_	?	2
16.2	430.3	C_27_H_43_NO_3_	Tetrahydrojervine *	2
16.5	574.3	C_33_H_51_NO_7_	Isomer of Cycloposine *	1, 2
16.6	618.4	C_35_H_55_NO_8_	?	2
16.9	578.4	C_33_H_55_NO_7_	?	2
17.1	428.3	C_27_H_41_NO_3_	Dihydrojervine * **	2
17.2	472.3	C_29_H_45_NO_4_	?	2
17.3	620.4	C_35_H_57_NO_8_	?	2
17.4	414.3	C_27_H_43_NO_2_	Etioline * **	2
17.5	428.3	C_27_H_41_NO_3_	Isomer of Dihydrojervine * **	2
17.6	410.3	C_27_H_39_NO_2_	Veratramine	2, 3
18.4/18.5	412.3	C_27_H_41_NO_2_	Cyclopamine	3, 4, 5
18.8	410.3	C_27_H_39_NO_2_	Isomer of Veratramine *	4
18.9	412.3	C_27_H_41_NO_2_	Isomer of Cyclopamine *	3, 4
19.2	474.3	C_29_H_47_NO_4_	?	4, 5
19.4	456.3	C_29_H_45_NO_3_	?	4, 5
19.6	416.3	C_27_H_45_NO_2_	22-keto-26-aminocholesterol *	5
19.5	414.3	C_27_H_43_NO_2_	Isorubijervine **	4
19.8	458.3	C_29_H_47_NO_3_	Muldamine	4, 5
20	416.3	C_27_H_45_NO_2_	Isomer of 22-keto-26-aminocholesterol *	5
20.3	398.3	C_27_H_43_NO	Verazine *	5
20.5	458.3	C_29_H_47_NO_3_	Isomer of Muldamine *	5

* Signifies a prediction based on the *m*/*z* and molecular formula. ** See Appendix A for data. ? Signifies an unknown compound that did not correlate with literature.

## Data Availability

Data supporting reported results can be found in the Appendix A.

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
