# Peer review of "Pharmacology of Veratrum californicum Alkaloids as Hedgehog Pathway Antagonists"

_pharmaceuticals, 2024, doi:10.3390/ph17010123_

Round 1
Reviewer 1 Report
Comments and Suggestions for Authors
The manuscript entitled Pharmacology of Veratrum californicum Alkaloids as Hedgehog 2 Pathway Antagonists addresses the identification and bioactivity evaluation of minor constituents obtained from extracts of roots and rhizomes of Veratrum californicum in regard to the inhibition Hh signaling.
The study is scientifically sound and represents a significant contribution for the scientists involved in the study of natural products as sources of lead compounds for drug discovery.
This work is composed according the scientific rules and contains all properly structures sections.
In the introductory section authors provide the background of study and indicate its principal objectives. Hedgehog Signaling Pathway has been adequately explained and illustrated. In addition the structures of naturally occurring and synthetic hedgehog signaling pathway inhibitors have been presented.
Materials and Methods section adequately explained the procedures of collecting and storing the plant material, extraction and separation procedures, and the analytical procedures and bioactivity testing.
In the results and discussion section authors clearly presented the study data in tabular and in graphical form. The obtained results were properly discussed and compared with published data. The references were properly cited.
In summary, in my opinion this manuscript should be accepted in present form.
Author Response
We thank the reviewer for the time to evaluate our work, and their kind words of support for the study.
Reviewer 2 Report
Comments and Suggestions for Authors
Abstract:
1. What is the software which was doing this? software was used to predict the molecular formulas
for every detected alkaloid.
2. What is the kind of this comparison? The raw extract and each fraction were compared to 0.1 µM cyclopa- mine (comparison of MWt, MF, 1D structure, 2D structure, biological activities…??
3. How could this happen? The Confirmation of the structure and suggestion of the isomer structure?? Via HPLC, HRMS…such confirmation of a 2D or 3D structure requires huge spectral work on pure compounds.
Introduction:
1. The purpose of the current work was to identify the minor constituents present in the raw extract that inhibit Hh signaling.
2. Fraction 1 is of particular interest as it had by far the greatest Hh inhibition with only four alkaloids present, two of which have been studied previously and have been reported to not sup- press Hh pathway signaling in this model system, implying the remaining two alkaloids
are responsible for the observed bioactivity.
3. Examination of chromatograms corresponding to fractions 1-5 (Figures S2-7) reveals four alkaloids in fraction 1, thirteen (including those from Figure S7) in fraction 2, three in fraction 3, seven (including one from Figure S7) in fraction 4, and eight in fraction 5. MS was used to identify veratrosine, cycloposine, and an isomer of each in fraction 1 (Figure S2). Cycloposine and an isomer, an isomer of veratrosine, tetrahydrojervine, dihydro- jervine and an isomer, etioline, veratramine, and five unknown alkaloids were found in Fraction 2 (Figures S3 and S7). Fraction 3 contained veratramine, cyclopamine, and an isomer of cyclopamine, which were all observed in other fractions (Figures S4 and S8). Fraction 4 contained isorubijervine, an isomer of veratramine, cyclopamine and an isomer, muldamine, and two unknown compounds (Figures S5 and S7). Finally, cyclopamine, muldamine and an isomer, 22-keto-26-aminocholesterol and an isomer, verazine, and two unknown compounds were present in fraction 5 (Figure S6) [4].
From the previous comments: it is not scientifically accepted to conclude the presence of metabolites in the most bioactive fraction according to two HRMS different in the third or fourth digit. Even if the HRMS are identical upto the sixth didgt we still need to isolate and purify the compound to confirm it structure with several spectral techniques such as NMR.
If the The purpose of the current work was to identify the minor constituents present in the raw extract that inhibit Hh signaling; then the main goal of this work is not done yet!
Author Response
Reviewer 2
Abstract:
Reviewer 2, Comment 1: What is the software which was doing this? software was used to predict the molecular formulas for every detected alkaloid.
Response to Reviewer 2, Comment 1: We have included the software in the abstract and materials and methods sections.
Abstract - Page 1, line 16: Mass spectrometry was used to identify the presence of twenty-five alkaloids, nine more than are commonly cited in literature reports, and the Bruker Compass Data Analysis software was used to predict the molecular formula for every detected alkaloid.
Materials and Methods - Page 7, lines 266-268: The mass and molecular formula for alkaloids were predicted using the Compass Data Analysis software (Bruker Corporation, Billerica, MA, USA).
Reviewer 2, Comment 2: What is the kind of this comparison? The raw extract and each fraction were compared to 0.1 µM cyclopamine (comparison of MWt, MF, 1D structure, 2D structure, biological activities…??
Response to Reviewer 2, Comment 2: We have added the statement, “Gli activity of the” in the abstract to address this comparison.
Abstract – Page 1, line 18: The sentence now reads: “The Gli activity of the raw extract and each fraction were compared to 0.1 µM cyclopamine, and fractions 1, 2, and 4 showed increased bioactivity through suppression of the Hh signaling pathway.” This sentence specifically refers to the bioactivity comparison portion of the paper.
Reviewer 2, Comment 3: How could this happen? The Confirmation of the structure and suggestion of the isomer structure?? Via HPLC, HRMS…such confirmation of a 2D or 3D structure requires huge spectral work on pure compounds.
Response to Reviewer 2, Comment 3: In each case where an alkaloid isomer is stated, the word “potential” has now been included. The rationale for the use of potential isomer terminology is that the two compounds have the same molecular formula as determined by HRMS, but different retention time by HPLC, and the suggested bioactivity of the potential isomers may be greater than the previously characterized alkaloid in select instances. The reviewer is correct, an exhaustive confirmation of structure was not performed, nor was it within the scope of the study conducted.
Introduction:
- The purpose of the current work was to identify the minor constituents present in the raw extract that inhibit Hh signaling.
- Fraction 1 is of particular interest as it had by far the greatest Hh inhibition with only four alkaloids present, two of which have been studied previously and have been reported to not suppress Hh pathway signaling in this model system, implying the remaining two alkaloids are responsible for the observed bioactivity.
- Examination of chromatograms corresponding to fractions 1-5 (Figures S2-7) reveals four alkaloids in fraction 1, thirteen (including those from Figure S7) in fraction 2, three in fraction 3, seven (including one from Figure S7) in fraction 4, and eight in fraction 5. MS was used to identify veratrosine, cycloposine, and an isomer of each in fraction 1 (Figure S2). Cycloposine and an isomer, an isomer of veratrosine, tetrahydrojervine, dihydro- jervine and an isomer, etioline, veratramine, and five unknown alkaloids were found in Fraction 2 (Figures S3 and S7). Fraction 3 contained veratramine, cyclopamine, and an isomer of cyclopamine, which were all observed in other fractions (Figures S4 and S8). Fraction 4 contained isorubijervine, an isomer of veratramine, cyclopamine and an isomer, muldamine, and two unknown compounds (Figures S5 and S7). Finally, cyclopamine, muldamine and an isomer, 22-keto-26-aminocholesterol and an isomer, verazine, and two unknown compounds were present in fraction 5 (Figure S6) [4].
Reviewer 2, Comment 4: From the previous comments: it is not scientifically accepted to conclude the presence of metabolites in the most bioactive fraction according to two HRMS different in the third or fourth digit. Even if the HRMS are identical upto the sixth didgt we still need to isolate and purify the compound to confirm it structure with several spectral techniques such as NMR.
Response to Reviewer 2, Comment 4: To address the reviewer’s comment, we have referred to each alkaloid of common molecular weight and predicted molecular formula, based on HRMS data, as a “potential” isomer. In a prior study, our lab invested two years into the evaluation of a single isomer to obtain the NMR data sufficient for definitive characterization of a potential isomer of cylcopamine. The concentration and stability of the isolated, purified, alkaloids posed challenges to the integrity of the NMR spectra. We have reached the end of our labs capacity to further characterize these compounds. Thus, if the reviewer has a preferred way for us to refer to these potential isomers, we welcome the feedback.
Reviewer 2, Comment 5: If the purpose of the current work was to identify the minor constituents present in the raw extract that inhibit Hh signaling; then the main goal of this work is not done yet!
Response to Reviewer 2, Comment 5: The word “identify” was replaced with “detect and evaluate.”
Introduction – Page 5, line191: The sentence now reads as follows: “The purpose of the current work was to detect and evaluate the minor constituents present in the raw extract that inhibit Hh signaling.” The detection was achieved through peak retention time in the HPLC chromatogram, and by use of HRMS. The evaluation was performed by way of bioactivity assessment using the Shh-Light II cell assay. Throughout the text, each time we refer to an alkaloid isomer, we have changed this to now refer to them as a “potential” isomer.
Reviewer 3 Report
Comments and Suggestions for Authors
Extracting and isolating natural products from plants, identifying their active components, and elucidating their biological functions are significant in drug development. The author extracted alkaloids from Veratrum californicum and discovered compounds with stronger antagonistic effects on the hedgehog signaling pathway. This provides candidate molecules for the development of hedgehog signaling pathway antagonists. However, the following issues exist in the study:
1. The author aimed to investigate the antagonistic effects of minor constituents alkaloids from Veratrum californicum on the hedgehog signaling pathway. Why did the author choose to differentiate compounds based on time intervals during the isolation process instead of using compound relative content?
2. It is well-known that natural products often have numerous isomers. Predicting compounds solely based on m/z for elemental composition may lead to errors. When predicting compounds, if standard substances were used, please provide data for these standards, along with the MS/MS fragment information of the detected compound.
3. In the GLI-Reporter activity analysis, the author mentioned treating cells with 0.05 and 0.01 μM cyclopamine, but the results do not reflect this. Why is that?
4. In the GLI-Reporter activity analysis experiment, the inhibitory effect of Fraction4 is significantly better than 0.1 μM cyclopamine. According to the relative content of alkaloids, the relative content of cyclopamine in Fraction4 is approximately 4 times higher than in 0.1 μM cyclopamine. Is this improved effect due to synergistic effects of other components or a dosage effect? The same question also arises in fraction1 and fraction2. The author should increase the dosage of cyclopamine to clarify the reasons for the better effect.
Author Response
Reviewer 3
Extracting and isolating natural products from plants, identifying their active components, and elucidating their biological functions are significant in drug development. The author extracted alkaloids from Veratrum californicum and discovered compounds with stronger antagonistic effects on the hedgehog signaling pathway. This provides candidate molecules for the development of hedgehog signaling pathway antagonists. However, the following issues exist in the study:
Reviewer 3, Comment 1: The author aimed to investigate the antagonistic effects of minor constituents alkaloids from Veratrum californicum on the hedgehog signaling pathway. Why did the author choose to differentiate compounds based on time intervals during the isolation process instead of using compound relative content?
Response to Reviewer 3, Comment 1: The raw extract HPLC-DAD chromatogram (Figure 3) consisted of a multitude of alkaloid peaks, tightly clustered together with similar retention times. We sought to obtain smaller numbers of alkaloids in each fraction to more easily detect and evaluate them. Several of these fractions were pooled in order to increase the abundance of each of the minor constituents for better detection. In response to the reviewer comment, we have added a sentence on page 8, lines 305-308 to described this: “These fractions were collected from several runs, pooled, dried, and reconstituted in order to concentrate the alkaloids for enhanced detection.”
Response to Reviewer 3, Comment 2: It is well-known that natural products often have numerous isomers. Predicting compounds solely based on m/z for elemental composition may lead to errors. When predicting compounds, if standard substances were used, please provide data for these standards, along with the MS/MS fragment information of the detected compound.
Response to Reviewer 3, Comment 2: With regard to the isomers, we have inserted the word “potential” in front of “isomer” each time it is used in the text. To address the comment on predicting compounds, we used the elution times and MS data for commercially available veratramine and cyclopamine standards for validation. On page 9, lines 340-344, we provide reference to these data that are available in the supplementary materials. “Etioline, dihydrojervine, and isorubijervine had been detected in V. californicum in the past, but in this work, they were not visible in the HPLC chromatograms. MS was used to identify these compounds by retention time, m/z ratios, and predicted molecular formulas corresponding to data from prior reports (see Figure S7) [4]. Validation of veratramine and cyclopamine by co-elution with standards is shown in Figure S8.”
Reviewer 3, Comment 3: In the GLI-Reporter activity analysis, the author mentioned treating cells with 0.05 and 0.01 μM cyclopamine, but the results do not reflect this. Why is that?
Response to Reviewer 3, Comment 3: This was an error on our part. Figure 4b, on page 10, has now been updated to show data for 0.01 and 0.05 µM cyclopamine.
Reviewer 3, Comment 4: In the GLI-Reporter activity analysis experiment, the inhibitory effect of Fraction 4 is significantly better than 0.1 μM cyclopamine. According to the relative content of alkaloids, the relative content of cyclopamine in Fraction 4 is approximately 4 times higher than in 0.1 μM cyclopamine. Is this improved effect due to synergistic effects of other components or a dosage effect? The same question also arises in fraction 1 and fraction 2. The author should increase the dosage of cyclopamine to clarify the reasons for the better effect.
Response to Reviewer 3, Comment 4: The recommendation of the reviewer has been explored in a prior study, citation 17, Turner, M. W.; Cruz, R.; Elwell, J.; French, J.; Mattos, J.; McDougal, O. M. Native V. Californicum Alkaloid Combinations Induce Differential Inhibition of Sonic Hedgehog Signaling. Molecules 2018, 23 (2222). https://doi.org/10.3390/molecules23092222.
The authors produced a cocktail of commercially available standards (cyclopamine, veratramine, isorubijervine, and muldamine) at commensurate concentration to those found in the raw extract, and the raw extract exhibited superior bioactivity. The conclusion of the study was that increased bioactivity was an additive effect from the additional alkaloids in the extract. These additional alkaloids were the minor constituents pursued in the current investigation. There is also a dosage effect occurring, as can be seen from the difference in bioactivity between the 0.1 and 0.01/0.05 µM cyclopamine standards (updated Figure 4b).
Round 2
Reviewer 2 Report
Comments and Suggestions for Authors
comment: line 204-206 (please add the ref. for your previous work. and if your work does not end in a publication please mention the major difficulty in the isolation and identification these compounds to show the hard work which have been done before.)
Fraction 1 is of particular interest as it had by far the greatest Hh
inhibition with only four alkaloids present, two of which have been studied previously
and have been reported to not suppress Hh pathway signaling in this model system, im-
plying the remaining two alkaloids are responsible for the observed bioactivity.
Author Response
Response to Reviewer Comment: A citation has been added for Matt Turner's dissertation [53] and his publication in Fitoterapia from 2019 [4]:
53. Turner, M. Comprehensive Investigation of Bioactive Steroidal Alkaloids in Veratrum Californicum, Boise State University, 2019.4. Turner, M. W.; Rossi, M.; Campfield, V.; French, J.; Hunt, E.; Wade, E.; McDougal, O. M. Steroidal Alkaloid Variation in Veratrum Californicum as Determined by Modern Methods of Analytical Analysis. Fitoterapia 2019, 137 (104281). https://doi.org/10.1016/j.fitote.2019.104281. In his dissertation work, Matt performed NMR experiments for the identification and characterization of the potential cyclopamine isomer, and in the Fitoterapia paper he resorted to HRMS for qualitative assessment of alkaloids. The process of isolating enough purified material for the NMR study was extremely tedious and time consuming, and the analyte did not remain stable for extended periods. As a result the HSQC NMR experiments gave some insights regarding the potential cyclopamine isomer, but the result was not definitive. A new sentence has been inserted at lines 206-209 that reads as follows: . "Nuclear magnetic resonance and high resolution mass spectrometry (HRMS) experiments were conducted for these two alkaloids, resulting in their potential identification as isomers of cyclopamine and veratramine [4,53]."